# The Potential of Using Cochayuyo (*Durvillaea incurvata*) Extract Obtained by Ultrasound-Assisted Extraction to Fight against Aging-Related Diseases

**DOI:** 10.3390/foods13020269

**Published:** 2024-01-15

**Authors:** Nicolás Muñoz-Molina, Javier Parada, Mario Simirgiotis, Romina Montecinos-González

**Affiliations:** 1Graduate School, Faculty of Agricultural and Food Sciences, Universidad Austral de Chile, Valdivia 5090000, Chile; nico94munoz@gmail.com; 2Institute of Food Science and Technology, Faculty of Agricultural and Food Sciences, Universidad Austral de Chile, Valdivia 5090000, Chile; romina.montecinos@alumnos.uach.cl; 3Instituto de Farmacia, Facultad de Ciencias, Universidad Austral de Chile, Valdivia 5090000, Chile; mario.simirgiotis@uach.cl

**Keywords:** *Durvillaea incurvata*, aging, enzyme inhibition, Alzheimer’s, diabetes, hypertension

## Abstract

The world’s population is in a demographical transition, with an increase in the number of older adults and prevalence of diseases related to aging. This study evaluated in vitro the potential of using *Durvillaea incurvata* extract (extracted using ultrasound-assisted extraction) to inhibit key enzymes associated with the development of age-related diseases. Our results show that an extract extracted via ultrasound-assisted extracted, as well as an extract conventional extracted from *Durvillaea incurvata,* presented antidiabetes potential by exhibiting inhibitory activity against α-glucosidase (91.8 ± 1.0% and 93.8 ± 0.3%, respectively, at 500 µg/mL) and α-amylase (42.2 ± 1.4% and 61.9 ± 0.9%, respectively, at 1500 µg/mL) enzymes related to starch digestion and postprandial glycemic response. Also, the extracts showed inhibitory activity against the enzymes acetylcholinesterase (51.5% and 50.8%, respectively, at 500 µg/mL) and butyrylcholinesterase (32.8% and 34.4%, respectively, at 0.5 mg/mL), the biomarkers associated with Alzheimer’s disease, and angiotensin-converting enzyme (98.7 ± 7.4% and 93.0 ± 3.4%, respectively, at 2.0 mg/mL), which is key in the regulation of vascular tone and blood pressure and helps to prevent the development of hypertension. In conclusion, the extract of *Durvillaea incurvata* obtained from ultrasound-assisted extraction has the potential to prevent the development of age-related pathologies such as diabetes, Alzheimer’s disease, and hypertension.

## 1. Introduction

Aging is a process characterized by the deterioration of the functional capacity of an organism, and its development is continuous, heterogeneous, universal, and irreversible [1]. During aging, there is a gradual reduction in homeostatic resilience, which is the ability to recover physiological parameters once they have been altered, and diseases can develop as a consequence of this [2]. These progressive changes are cumulative and increase the incidence of diseases such as diabetes, hypertension, and Alzheimer’s. One of the most accepted theories to explain aging corresponds to the oxidative stress theory of aging, which postulates that this is the result of the organism being inadequate protected against damage induced by free radicals, also called reactive oxygen and nitrogen species (RONS) [3]. This imbalance between the production of free radicals and the body’s antioxidant defenses generates oxidative stress, the accumulation of which throughout life plays a fundamental role in the pathogenesis of many diseases and aging [4]. In our world, the elderly population is constantly expanding, with a consequent increase in the prevalence of diseases related to aging. Therefore, bioactive compounds of natural origin with antioxidant capacity have received interest and been proposed to reduce the development of several aging-related diseases.

Brown algae (pheophytes) are a large and diverse group of organisms that comprise around 2000 species and are distributed in multiple marine ecosystems, presenting complex multicellularity and a wide morphological diversity among species [5]. In Chile, there is the *Durvillaea incurvata* seaweed, an endemic species that is known under the name of cochayuyo, which is commonly collected for human consumption [6]. It is known that this seaweed has high contents of proteins, essential amino acids, vitamins, and dietary fibers [7], but recently, it has also been demonstrated to be a rich source of bioactive compounds such as phlorotannins and fucoxanthin while having antioxidant activity and other healthy properties including antihyperglycemic, antiobesity, and neuroprotective activities [8,9,10].

Regarding the extraction of bioactive compounds from natural sources such as cochayuyo, in recent years, conventional extraction has been considered inefficient due to its time-consuming nature, high cost, and the degradation of the quality of the samples involved, whereas another method, namely ultrasound-assisted extraction, has been found to be more efficient due to its low energy requirement and efficiency with respect to time and solvent consumption. The ultrasound helps the solvent to penetrate the cells by destroying their cell walls, thus increasing the overall efficiency of the process [11]. Although there is plenty of evidence on the effects of using ultrasounds in extraction processes, research on the specific properties of these extracts is still limited for species such as cochayuyo, while the existing research points towards the potential of using brown seaweed to obtain healthy bioactive extracts, namely phlorotannins, which have demonstrated several biological activities, including antioxidant, anticancer, anti-inflammatory, antimicrobial, antidiabetic, antiviral, and antiallergy activities [12,13,14,15]. Thus, the main objective of this study was to evaluate the ability of an extract of *Durvillaea incurvata* (obtained using ultrasound-assisted extraction) to inhibit key enzymes in the development of aging-related diseases such as diabetes, Alzheimer’s disease, and hypertension, providing the bases for the further development of healthy food ingredients.

## 2. Materials and Methods

### 2.1. Chemicals and Reagents

All chemicals and reagents were of analytical grade. Most of them, including the Folin–Ciocalteu phenol reagent, 2,2-diphenyl-1-picrylhydracil (DPPH), and enzymes we used, were acquired from Sigma Chemical Co. (Saint Louis, MO, USA) unless stated otherwise.

### 2.2. Seaweed Sample Collection and Preparation

Cochayuyo seaweed (*Durvillaea incurvata*) was collected from the “Palo Muerto” sector (Latitude: −39.8833 Longitude: −73.5167) of Southern Chile, cleaned with seawater, and transported to the lab all in the same day. Once in the lab, the algae were washed with distilled water, cut into cubes of ~1 cm^3^, frozen at −80 °C, lyophilized, ground to a size of ~0.05 mm, and finally stored at −80 °C until extraction.

### 2.3. Optimization of Ultrasound-Assisted Extraction

The response surface methodology (RSM) was used to optimize the ultrasound-assisted ethanolic extraction. The extraction procedure was adapted from Dang et al. (2017) [11]. Ethanol/water 70% *v*/*v* and an ultrasonic processor (Sonics VCX series, 500 W, 20 kHz, Sonics & Materials Inc., Newtown, CT, USA) equipped with a titanium alloy (Model 208-B) probe (19 mm diameter) were used. An amplitude of 50% was employed for ultrasonic extraction. The solvent/dehydrated seaweed ratio was 50 (mL g^−1^). A Box–Behnken experimental design was used (see Table 1), and the independent variables were the extraction temperature (*X*_1_; 30–50 °C; maintained by using a thermoregulated water bath), extraction time (*X*_2_; 30–90 min), and ultrasound pulse cycle (*X*_3_; 8–12 s) (cycle-off = pulse cycle), while the response variables were the total phenolic content (*Y*_TPC_) and the antioxidant activity (*Y*_DPPH_ and *Y*_ORAC_, for DPPH and ORAC, respectively). Immediately after extraction, each extract was filtered with a 0.45 µm cellulose syringe filter and stored at −80 °C until analysis. Quadratic models (excluding less significant effects) were used for each response. Multiple response optimization was performed by using the “Desirability” function. All bioactivity-related analyses were performed on this optimized extract.

As a control, conventional extraction (CE) was performed using the following parameters: ethanol/water 70% (*v*/*v*), temperature 30 °C, agitation 60 rpm, and extraction time 12 h. The extract obtained through conventional extraction was also filtered through a 0.45 μm cellulose syringe filter and stored at −80 °C until analysis. For this extract, the solvent/dehydrated seaweed ratio was also 50 (mL g^−1^).

#### 2.3.1. Total Phenolic Content

The total phenolic content was assessed by the Folin–Ciocalteu (FC) method using gallic acid as a standard to construct the calibration curve (results expressed in mg of gallic acid equivalent, GAE, per 100 g d.w.) [16]. In brief, 0.5 mL of the sample or solvent blank was diluted in 3.75 mL of distilled water. Afterward, 0.25 mL of the FC reagent was added and homogenized. Then, 0.5 mL of the sodium carbonate solution (10% *w*/*v*) was added. The resulting solution was homogenized and incubated for 1 h at room temperature in the darkness. The absorbance of the reaction product was measured at 765 nm (UV spectrophotometer 1240, Shimadzu, Kyoto, Japan). Analyses were performed in triplicate.

#### 2.3.2. Antioxidant Activity

The antioxidant activity was measured by using two assays: DPPH and ORAC.

The antiradical activity, 2,2-diphenyl-1-picrylhydracil (DPPH), was measured by using the method of Tierney et al. [17]. First, a working solution of DPPH (0.048 mg/mL) was prepared by diluting a stock (0.238 mg/mL in methanol). For the analysis, 0.5 mL of DPPH solution was added to microtubes with 0.5 mL of the extract. After homogenization, the tubes’ contents were subjected to a reaction for 30 min at room temperature, and the absorbance was measured at 520 nm on a UV 1240 spectrophotometer (Shimadzu, Kyoto, Japan). Trolox was used as the reference standard. The results were expressed in µmol equivalent of Trolox (ET)/g 100 g dry seaweed (µmol ET/100 g d.w.). Analyses were performed in triplicate.

As said before, the ORAC method was also used to measure antioxidant activity. The reaction was carried out in a 75 mM phosphate buffer (pH 7.4) in a 96-well microplate. A total of 45 µL of the sample and 175 µL of fluorescein 108 mM were deposited. This mixture was incubated for 30 min at 37 °C; after that time, 50 µL of the AAPH solution 108 mM was added. The microplate was immediately placed in a dual-scan microplate spectrofluorometer (Gemini XPS, San Jose, CA, USA) for 60 min; fluorescence readings were recorded every 3 min (wavelengths of 485 nm excitation and 535 nm emission). The microplate was automatically shaken before and after each reading. For the calibration curve, Trolox was used at 6, 12, 18, and 24 M. All reactions were carried out in triplicate. The area under the curve (AUC) for each sample was calculated by integrating the relative fluorescence curve (r^2^ > 0.99). The net AUC of the sample was calculated by subtracting the AUC of the blank. The regression equation between the net AUC and Trolox concentration was determined, and the ORAC values were expressed as µmol Trolox equivalents/100 g of dry seaweed (µmol ET/100 g d.w.) using a previously established standard curve [18].

### 2.4. Inhibition of α-Glucosidase and α-Amylase Enzymes

The ability of the extracts to inhibit the α-glucosidase activity was measured using the method described by Nampoothiri et al. [19] and subsequently adapted by Lordan et al. [20]. Briefly, 50 µL of 100 mM extract in sodium phosphate buffer (pH 6.9) and 50 µL of 5 mM p-nitrophenyl-α-D-glucopyranoside in phosphate buffer were mixed in a 96-well microplate and incubated at 37 °C for 5 min. Then, 100 µL phosphate buffer was added to each well, which contained 0.1 U/mL α-glucosidase. A microplate reader set at 37 °C was used to record absorbance at a wavelength of 405 nm for 30 min. Blank (no enzyme) readings were subtracted from each well. The inhibitory effects of the extracts are expressed as IC_50_ values, which refer to the concentration that inhibits 50% of the enzyme activity. The pharmacological inhibitor, acarbose, was included as a positive control. The activity of α-glucosidase was calculated as follows:(1)Inhibition  (%)=(1−extract absorbance/control absorbance)×100
where the control is the enzyme–substrate reaction in the absence of inhibitors.

The potential of the extracts to inhibit the activity of α-amylase was also measured using the method described by Nampoothiri et al. (2011) and subsequently adapted by Lordan et al. (2013) [19,20]. A volume of 100 µL of extract and 1% starch solution in 20 mM sodium phosphate buffer was taken (pH 6.9 with 6 mM sodium chloride) and kept in Eppendorf tubes at 25 °C. A 100 µL volume of porcine pancreatic α-amylase (0.5 mg/mL) was added to each tube and then incubated at 25 °C for 10 min. The reaction was stopped by adding 200 µL of dinitrosalicylic acid reagent and incubating the tubes at 100 °C for 5 min. The samples were cooled to room temperature, and then 50 µL was taken from each tube and transferred to the wells of a 96-well microplate. The mixture was diluted by adding 200 µL of water to each well, and the absorbance was measured at a wavelength of 540 nm. Blank (no enzyme) readings were subtracted from each well. The inhibitory effects of the extracts are expressed as IC_50_ values, and acarbose was also included as a positive control. The α-amylase activity was also calculated using Equation (1).

### 2.5. Inhibition of the Acetylcholinesterase and Butyrylcholinesterase Enzymes

The inhibitory activity of the extracts against cholinesterase enzymes was evaluated as described by Ellman [21]. Briefly, 5-dithio-bis(2-nitrobenzoic) acid (DTNB) was dissolved in Tris-HCl buffer (pH 8.0) containing NaCl 0.1 M and MgCl_2_ 0.02 M. Then, the filtered was extract dissolved in deionized water (50 mL, 2 mg/mL), mixed in a 96-well microplate with 125 mL of DTNB, acetylcholinesterase (AChE), or butyrylcholinesterase (BChE) solution (25 mL) dissolved in Tris-HCl buffer (pH 8.0), and incubated for 15 min at 25 °C. The reaction was started by the addition of acetylthiocholine iodide (ATCI) or butyrylthiocholine chloride (BTCl) (25 mL). In addition, a blank was prepared by adding the solution sample to all reagents without the enzyme solutions (AChE or BChE). After 10 min of reaction, absorbance was measured at a wavelength of 405 nm. Finally, the IC_50_ (µg/mL) values were determined.

### 2.6. Inhibition of Angiotensin-I Converting Enzyme

The enzyme activity inhibition assay was carried out as described by Hou et al. (2003), modified by Jung et al. (2006) [22,23]. N-[3-(2-furyl)acryloyl]-Phe-Gly-Gly (FAPGG) (0.5 mM) and various concentrations of samples were completely dissolved in 50 mM Tris-HCl buffer (pH 7.5). Next, 20 µL of angiotensin-converting enzyme (ACE-I; 1 U/mL dissolved in 50 mM Tris-HCl buffer) was mixed with 200 µL of samples of various concentrations or with 50 mM Tris-HCl buffer (negative control). Then, 1 mL of FAPGG (0.5 mM) was added to the reaction mixture, and the absorbance was measured at 345 nm wavelength at 0, 5, 30, and 60 min. Captopril (antihypertensive agent) was used as a positive control. The inhibition value was calculated using the following equation:Inhibition (%) = (1 − [Absorbance at 60 min − Absorbance at 0 min]/[Control absorbance at 60 min − Control absorbance at 0 min]) × 100(2)

### 2.7. Statistics

For extraction optimization, experiments and data analyses were performed by the using response surface methodology (RSM) and STATGRAPHICS Centurion XV software, version 15.2.06 (Old Tavern Rd, The Plains, VA, USA), considering a level of confidence of 95%. For any means comparison, data were analyzed by conducting an analysis of variance (ANOVA) followed by Tukey’s Multiple Comparison test (*p* < 0.05). The same software (STATGRAPHICS) was used for all analyses.

## 3. Results

### 3.1. Optimization of Ultrasound-Assisted Extraction

For ultrasound-assisted ethanolic extraction optimization using the RSM, the Box–Behnken experimental design was ran, and the results are shown in Table 1. For each independent variable (total phenolic content, and antioxidant activity assessed by two methods), polinomial equations were fitted by excluding the less significant effects. Fitted equations, having the highest adjusted determination coefficient (R^2^-adjusted), are shown in Equations (3)–(5). For *Y*_TPC_, R^2^ was 68.6%, while R^2^-adjusted was 51.1%. For *Y*_DPPH_, the same values were as follows: R^2^ = 72.1% and R^2^-adjusted = 51.17%. Finally, for *Y*_ORAC_, R^2^ and R^2^-adjusted were 37.3% and 20.2%, respectively. These values show how capable the models are with respect to explaining data variability. 

Through using the multiple optimization procedure, the optimal conditions for extraction were obtained (goal: maximize *Y*_TPC_, *Y*_DPPH_, and *Y*_ORAC_). These conditions and theorical optimal responses are also shown in Table 1, while a comparison between the experimental results obtained at the optimal conditions and those obtained via conventional ethanolic extraction is shown in Table 2. Our results show that the extract obtained by ultrasound-assisted ethanolic extraction at optimal conditions has a similar content of phenolic compounds to the conventional extract but a higher antioxidant activity (*p* < 0.05).
*Y*_TPC_ = 7584.35 + 8.96*X*_1_ − 23.06*X*_2_ − 1244.21*X*_3_ + 2.05*X*_2_*X*_3_ + 58.08*X*_3_^2^(3)
*Y*_DPPH_ = −1002.91 + 37.80*X*_1_ + 29.63*X*_2_ + 374.78*X*_3_ − 3.87*X*_1_*X*_3_ − 2.47*X*_2_*X*_3_ − 4.46*X*_3_^2^(4)
*Y*_ORAC_ = 15,679.6 + 441.19*X*_1_ + 80.08*X*_2_ − 171.84*X*_3_(5)

### 3.2. Inhibition of α-Glucosidase and α-Amylase

Figure 1 shows how the activities of the enzymes α-glucosidase and α-amylase were affected by UAEoc, CE, and acarbose. Figure 1a shows that as the concentration of UAEoc, CE, and acarbose increased (10–500 µg/mL), the inhibition of the activity of the α-glucosidase increased. At the highest concentration (500 µg/mL), UAEoc, CE, and acarbose generated 91.8 ± 1.0%, 93.8 ± 0.3%, and 35. 9 ± 3.3% of inhibition, respectively. The IC_50_ values for the inhibition of α-glucosidase activity were 155 ± 16, 94 ± 18, and 642 ± 58 µg/mL for UAEoc, CE, and acarbose, respectively. The results (inhibition at highest concentration and IC_50_) indicate that were no differences between UAEoc and CE while demonstrating that seaweed extracts are more efficient than acarbose (*p* < 0.0001) in terms of α-glucosidase inhibition.

On the other hand, Figure 1b shows that, in the tested range (250–1500 µg/mL), acarbose inhibited α-amylase at a constant level (~60%), while UAEoc and CE increased their levels of inhibition with increasing concentration, reaching 42.2 ± 1.4% and 61.9 ± 0.9% inhibition, respectively. The IC_50_ values for α-amylase were 1680 ± 71 µg/mL, 1048 ± 29 µg/mL, and 144 ± 2 µg/mL, for UAEoc, CE, and acarbose, respectively, and all these values are statistically different (*p* < 0.0001), meaning that acarbose has the highest inhibition capacity, followed by CE and, finally, UAEoc.

### 3.3. Inhibition of the Enzymes Acetylcholinesterase and Butyrylcholinesterase

Figure 2 shows the inhibition of AChE and BChE enzymes in the presence of UAEoc and CE at increasing concentrations. The results show that as CE and UAEoc increased (0.01–500 µg/mL), the inhibition of AChE increased from ~44% to ~51%, with no other differences being observed at any other concentration (*p* > 0.05) (Figure 2a). The IC_50_ values were 48.55 ± 0.021 µg/mL for UAEoc and 153.15 ± 0.029 µg/mL for CE. Therefore, both extracts can inhibit the activity of AChE.

Regarding the effect on BChE, Figure 2c shows that the extracts are also capable of inhibiting this enzyme, with inhibition depending on concentration. Approximately 34% inhibition at 500 µg/mL (highest tested concentration) was achieved by both extracts (UAEoc and CE) (*p* > 0.05). The IC_50_ values were 87.58 ± 0.044 µg/mL for UAEoc and 121.79 ± 0.071 µg/mL for CE. Galantamine, a commercial inhibitor of cholinesterase enzymes and a drug used in treatments for Alzheimer’s, was used as a positive control. The IC_50_ values for this commercial inhibitor were 0.266 ± 0.029 µg/mL and 3.824 ± 0.025 µg/mL for AChE and BChE, respectively, which means that the standard drug galantamine is more efficient with respect to inhibiting these enzymes than the cochayuyo extracts.

### 3.4. Inhibition of Angiotensin-I-Converting Enzyme (ACE)

The activity of ACE was affected by the cochayuyo extracts (100–2000 µg/mL) and positive control Captopril (pharmacological inhibitor), as shown in Figure 3, which depicts the inhibition percentages of the cochayuyo extracts and Captopril. Both extracts, UAEoc and CE, inhibited ACE in a concentration-dependent way (Figure 3a). At the highest extract concentration (2000 µg/mL), UAEoc inhibited the enzyme’s activity until 98.7 ± 7.4%, while CE achieved 93.0 ± 3.4%. This inhibition capacity was lower than that generated by Captopril, which produced 95.0 ± 2.1% of inhibition at 100 ng/mL (Figure 3b). The IC_50_ values were 613.951 ± 80.169 µg/mL, 901.219 ± 40.611 µg/mL, and 6.810 × 10^−3^ ± 1.379 × 10^−3^ µg/mL for UAEoc, CE, and Captopril, respectively. In general, no statistically significant differences were observed between UAEoc and CE regarding inhibition at the highest concentration and the IC_50_ values; however, significant differences in this regard were observed between the extracts and Captopril.

## 4. Discussion

### 4.1. Optimization of Ultrasound-Assisted Extraction

The optimal conditions for extraction were achieved by using the RSM, although the determination coefficients (between 37.3 and 72.1%) were relatively lower than those obtained by other authors, such as Dang et al. [11] (R^2^ > 90%), Mohamed Ahmed et al. [24] (R^2^ > 80%), and Vuong et al. [25] (R^2^ between 53 and 88%). This could mean that the variability of the process is high or that the “real” optimum conditions for extraction are beyond the experimental range of this study. Nevertheless, the optimized extraction was more efficient than the conventional method, since the extract showed higher antioxidant activity (Table 2). Given that no differences were found regarding total phenolic compounds (despite antioxidant activity), there is a possibility the that phenolic profiles could be different, or that the ultrasound-assisted method is capable of extracting compounds other than phenolics, such as tocols (tocopherols and tocotrienols), which are abundant in cochayuyo and also contribute to antioxidant activity [25]. Recently, the RSM was used to optimize the extraction of compounds with antioxidant and neuroprotective activities from cochayuyo, but this study involved using pressurized liquids, and their results showed that the optimal conditions for extraction were 180 °C and a water/ethanol ratio of 71:29 [10]. Moreover, in another study involving the use of pressurized liquids (50% ethanol at 120 °C and 1500 psi), the extracts showed antihyperglycemic capacities [8], confirming that bioactive compounds can be successfully extracted from this type of seaweed. In general, following different methods can lead to different results regarding optimum extraction conditions.

### 4.2. Inhibition of α-Glucosidase and α-Amylase

The results obtained are consistent with those described in previous investigations.

Regarding algae’s capacities to inhibit the activity of the enzymes α-glucosidase and α-amylase, Erpel et al. (2021) showed that an extract of phlorotannins obtained from *Durvillaea incurvata* from Niebla at a concentration of 500 µg/mL inhibited the activity of the α-glucosidase enzyme by approximately 80% with an IC_50_ of 245.1 ± 5.3 µg/mL and the activity of acarbose around 40% with an IC_50_ of 659.5 ± 36.7 µg/mL. Regarding α-glucosidase inhibition, the present study’s extracts showed a slightly higher percentage inhibition (at same concentration) and IC_50_ values lower than the extract described by Erpal et al. (2021), thus indicating that they have a relatively higher inhibitory capacity. On the other hand, regarding α-amylase inhibition, these authors reported no effect on the enzyme’s activity, while our extracts do demonstrate inhibition capacities. This may be since different extraction methods were used (pressurized hot liquid vs. ultrasonic assisted), which may generate a different profile of bioactive compounds with different inhibition capacities [26]. Another study reported that ethanolic and acetone extracts of cochayuyo, at a concentration of 1000 µg/mL, inhibited α-glucosidase by 96.9 ± 0.4 and 99.3 ± 0.3%, showing IC_50_ values of 473.4 ± 0.9 and 466.0 ± 1.3 µg/mL, respectively (acarbose 797.85 ± 1.1 µg/mL) [12]. Based on the IC_50_ values, the extracts investigated in the present study appear be more efficient with respect to inhibiting this enzyme than those reported previously.

Regarding α-amylase, it has been reported that the inhibitory effect of cochayuyo extracts depends on the extraction method used, with extracts derived from acetonic extraction (43.4 ± 2.0% inhibition at 2000 µg/mL) being more efficient than those obtained from ethanolic extraction (0% inhibition) [12]. The present study’s outcomes suggest that UAEoc (but also CE) is adequate for generating an antihyperglycemic ingredient, especially considering that a high inhibition of α-glucosidase, along with a moderate inhibition of α-amylase, would be better, since it could avoid some unwanted side effects related to excessively digested amounts of starch reaching the colon [20] and also because it has been reported that a high α-amylase activity at the oral level would be associated with improved glycemic homeostasis (lower glycemic response is achieved) following starch ingestion due to early insulin release [27].

### 4.3. Inhibition of the Enzymes Acetylcholinesterase and Butyrylcholinesterase

Regarding algae’s capacity to inhibit the activity of the AChE and BChE enzymes, Nho et al. (2020) previously evaluated the neuroprotective effects of a Phlorotannin-rich extract derived from *Ecklonia cava* (PEEC), an edible brown alga. In this study, PEEC (1000 μg/mL) achieved 95.4 and 74.7% inhibition of AChE and BChE, respectively, which means that PEEC has a higher inhibitory capacity than our extracts (see Figure 2), probably due to the different concentrations and profiles of phlorotannins [13]. 

Another study evaluated the anticholinesterase potential of hydroethanolic extracts derived from some South African marine algae, namely *Ecklonia maxima* (ECK), *Gelidium pristoides* (GLD), *Gracilaria gracilis* (GCL), and *Ulva lactuca* (ULT) [28]. At 500 μg/mL, the inhibition of AChE was approximately 15% for ULT, 20% for GLD, and 25% for GCL and ECK, which is lower than the inhibition achieved by the extracts investigated in our study at the same concentration (see Figure 2a). This lesser capacity may be due to the profiles phlorotannins or the fact that the extraction method used was not optimized to maximize the extraction of polyphenols, unlike that used by our group. On the other hand, at the same concentration (500 μg/mL), the inhibition of the BChE was approximately 20% for ULT, 25% for GLD, and 30% for GCL and ECK, and these values are similar to the inhibition rates achieved by our extracts (UAEoc and CE) (see Figure 2b). 

### 4.4. Inhibition of Angiotensin-I-Converting Enzyme (ACE)

The potential of using brown seaweed as a antihypertensive agent due to its capacity to inhibit ACE has been previously noted. For instance, Shih et al. (2022) analyzed the inhibition achieved by extracts obtained by enzymatic extraction from *Durvillaea antarctica* [14]. Said extracts (1000 µg/mL)—Dur-A, Dur-B, and Dur-C—generated ACE activity inhibition values of 72.5 ± 1.4%, 80.7 ± 1.6%, and 62.9 ± 0.6%, respectively. At the same concentration (1000 µg/mL), UAEoc generated an inhibition of ACE similar to Dur-B, while CE achieved a lower level of inhibition than the three extracts (see Figure 3a). These outcomes suggest that UAEoc has greater potential with respect to ACE inhibition.

## 5. Conclusions

The results of this study show that ultrasound-assisted extraction is more efficient than conventional extraction, especially when one is aiming to optimize the antioxidant activity of their extracts. The cochayuyo extracts (both UAEoc and CE) presented inhibitory activities on the enzymes α-glucosidase and α-amylase which were even higher than the inhibition demonstrated by the positive control we used, showing the potential to prevent postprandial hyperglycemia and the development of related diseases such as diabetes. The extracts also showed inhibitory activity on AChE and BChE enzymes at levels comparable with inhibitors obtained from other natural sources, exhibiting potential for use in treatments aiming to fight and/or protect against Alzheimer’s disease. Regarding antihypertensive potential, the extracts showed inhibitory activities on ACE, an enzyme that plays a key role in regulating vascular tone and blood pressure, suggesting that these extracts could help to prevent hypertension. So, this study’s results show that cochayuyo hydroethanolic extracts have potential for using in edible products designed to fight against aging-related diseases such as diabetes, Alzheimer’s disease, and hypertension. Further research is needed to study the incorporation of these extracts in foods and corroborate their effects in vivo.

## Figures and Tables

**Figure 1 foods-13-00269-f001:**
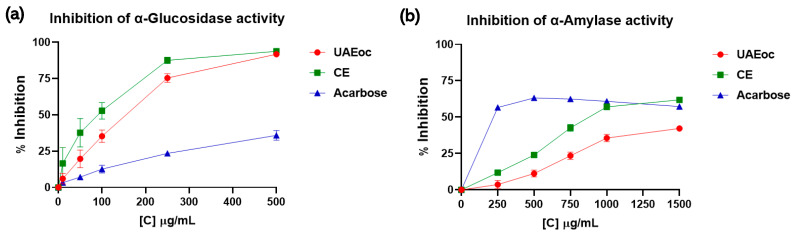
The inhibitory effects of the cochayuyo extracts (UAEoc and CE) on amylolytic enzymes. (**a**) Effect on α-glucosidase. (**b**) Effect on α-amylase. Each point represents the average of three measurements.

**Figure 2 foods-13-00269-f002:**
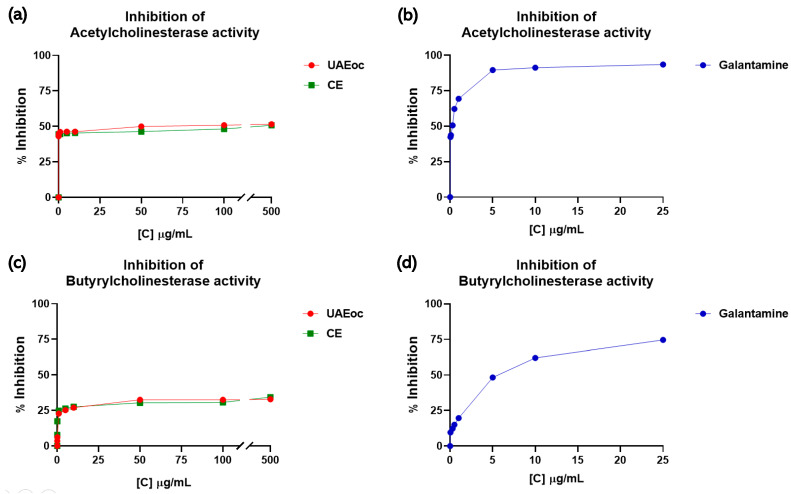
The inhibitory effects of the cochayuyo extracts (UAEoc and CE) and galantamine on cholinesterases enzymes. (**a**,**b**) Effect on AChE; (**c**,**d**) Effect on BChE. Each point represents the average of three measurements.

**Figure 3 foods-13-00269-f003:**
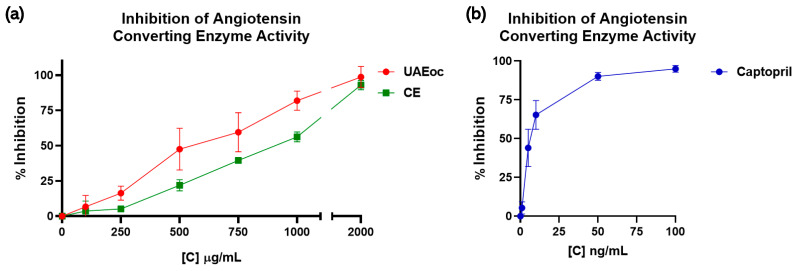
Inhibitory effects of the cochayuyo extracts (UAEoc and CE) on ACE activity (**a**) and the inhibitory effect of Captopril (**b**).

**Table 1 foods-13-00269-t001:** Total phenolic content and antioxidant activity values (obtained using both DPPH and ORAC assays) for the optimization of the ultrasound-assisted extraction using the RSM and a Box–Behnken experimental design.

Run	Temperature (°C)	Time (min)	Pulse Cycle (s)	TPC (mg GAE/100 g d.w.)	DPPH (µmol ET/100 g d.w.)	ORAC (µmol ET/100 g d.w.)
1	40	60	10	1330.5 ± 152	2628.45 ± 252	36,215.79 ± 6410
2	30	30	10	955.5 ± 199	2513.12 ± 201	28,164 ± 6030
3	50	30	10	1318 ± 120	2275.05 ± 210	33,089.06 ± 4523
4	30	90	10	1065.5 ± 149	2758.27 ± 112	27,259.37 ± 1538
5	50	90	10	1155.5 ± 134	2641.62 ± 280	39,037.26 ± 2495
6	30	60	8	1265.5 ± 231	2445.65 ± 160	33,212.75 ± 2634
7	50	60	8	1413 ± 145	2742.52 ± 144	43,489.53 ± 6475
8	40	60	10	949.66 ± 233	2426.61 ± 203	48,317.53 ± 6973
9	30	60	12	1321.33 ± 230	2439.32 ± 130	37,028.04 ± 5377
10	50	60	12	1438 ± 142	2426.76 ± 170	45,343.35 ± 5878
11	40	30	8	1538 ± 83	2267.95 ± 210	34,163.16 ± 5366
12	40	90	8	1013 ± 115	2851.92 ± 148	41,732.59 ± 4721
13	40	30	12	1575.5 ± 210	2586.98 ± 165	30,435.01 ± 4468
14	40	90	12	1543 ± 150	2578.56 ± 196	37,042.25 ± 2750
15	40	60	10	1318 ± 146	2679.03 ± 155	31,677.9 ± 3752
Optimal	50.0	80.8	8.0	1258.8 *	2851.0 *	42,834.0 *

Experimental outcomes are shown as mean ± standard deviation. d.w. means dry weight. * Theoretical values at optimal conditions, according to multi-response optimization analysis.

**Table 2 foods-13-00269-t002:** Comparison of the extracts obtained from ultrasound-assisted extraction at optimal conditions (UAE_OC_) and conventional extraction (CE).

Extract	Temperature (°C)	Time (min)	Pulse Cycle (s)	TPC (mg EAG/100 g d.w.)	DPPH (µmol ET/100 g d.w.)	ORAC (µmol ET/100 g d.w.)
UAE_OC_	50.0	80.8	8.0	1280.0 ± 225 a	2550.8 ± 205 a	36,274.3 ± 6250 a
CE	30.0	720 *	-	1178.0 ± 150 a	1589.38 ± 63 b	27,219.9 ± 2100 b

* 12 h at 60 rpm agitation. Values are means ± standard deviations (*n* = 3). The presence of different letters in the same column indicates a significant difference (*p* < 0.05).

## Data Availability

Data is contained within the article.

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
