# Peer review of "The Potential of Using Cochayuyo (Durvillaea incurvata) Extract Obtained by Ultrasound-Assisted Extraction to Fight against Aging-Related Diseases"

_foods, 2024, doi:10.3390/foods13020269_

Round 1

Reviewer 1 Report

Comments and Suggestions for Authors

Introduction:

Despite of being well organized, in generally the information is too vague. I was expecting to see more information/details namely about the sample in study. Only 4 lines (46 to 50) report something about the sample, I suggest including references from other studies, and more details about the characterization and properties of the Cochayuyo, if available in literature.

Materials and methods:

Its missing a section with reagents.

The information from sample collection and preparation (lines 72 to 76) should be separated from the rest of the information in section “Optimization of ultrasound-assisted extraction”.

How do ensure that some temperatures were maintained constant during the long periods of extraction, for example 50 ºC during 90 min? how do you control the temperature?

Why in the ORAC assay were performed triplicates and in the other two assays, TPC and DPPH, only duplicates?

Its missing the section explaining how statistical analysis was made.

Results:

Line 199 correct the word compound.

Table 1, correct Total phenolic compounds to Total phenolic content.

Table1, The units from TPC, DPPH and ORAC assays are all xxx/100 g dw, but in the materials and methods section is reported that the results will be presented in xxx/g dw please correct in the section 2.

Table 1, why did you not calculate the SD associated with the measurements? This information should be added to each individual value for each condition tested.

When RSM is used to optimize the extraction conditions a more detailed information and statistical analysis should be provided. In fact, it is missing a comparison/validation of the proposed method by comparing the results obtained by the software for the optimal extraction conditions and the experimental results when applying the predicted optimal conditions? Why is this comparison and statistical analysis not made?

Figure 2, why Galantamine was separated from the other extracts? In figure 1 they were all in the same graphic.

Discussion:

Section 4.1, It is missing a comparison with literature. No comparison and no discussion of the obtained results is made, the authors only limit to report the data.

It is missing also HPLC analysis, which in my opinion is crucial to see which type of compounds are contributing to the exhibited properties.

Reviewer 2 Report

Comments and Suggestions for Authors

The topic of the manuscript is pertinent and offers intriguing insights for readers. The manuscript is well structured. The introduction outlines well the background and significance of the study. The methods employed are appropriate and clearly explained. The results are effectively presented and accompanied by relevant references in the discussion. While the manuscript presents compelling findings from a lab-scale study, it requires revisions to enhance clarity and completeness.

1.     Page 2, line 62. Please review and correct the references to ensure accuracy. After reference 10, the next citation should be reference 11, not 19.

2.     There is no information provided regarding the purity of the substances used in the analyses or their sources of purchase. For instance, details about the purity and sources of the FC reagent, gallic acid standard, and sodium carbonate for TPC determination are missing. This lack of information applies to other assays as well.

3.     Page 2, subchapter 2.1 requires improvement. There is a lack of information regarding the ultrasound power or amplitude employed in the extraction process. The solvent mentioned is 70% ethanol (v/v), but clarification is needed—was this a mixture of ethanol and water, or ethanol combined with another solvent? Additionally, details about the pulse cycle of sonication, such as the duration of the off cycle, are absent. While a stirring rate of 60 rpm was specified for CE, there is no information provided for UAE. Were the ultrasound extractions conducted without stirring the mixture? Furthermore, the ratio of solvent to plant material for both CE and UAE is undisclosed.

4.     Line 94. The results are expressed in milligrams of gallic acid equivalents per gram of gallic acid equivalents or per gram of dry plant material? The parentheses need to be rephrased for clarity.

5.     Line 108. Please revise the following phrase: "After homogenizing, the tubes reacted for 30 min at room temperature, and the absorbance was measured at 520 nm on a UV 1240 spectrophotometer (Shimadzu, Kyoto, Japan)." The tubes themselves cannot react.

6.     Line 130. Reference 11 appears for the first time after reference 22. Kindly review and correct the order of all references and citations throughout the manuscript to ensure proper sequencing and citation consistency.

7.     Please revise and standardize the usage of measurement units throughout the manuscript. For instance, in Subchapter 2.3, ensure uniform spelling and formatting of units, such as 'minutes.'

8.     Table 1. Please standardize the units of TPC. In Subchapter 2.1, it is referred to as GAE, while in Table 1, it is mentioned as EAG. Additionally, please provide an explanation for the abbreviation 'd.w.' as it is not currently elucidated in the Material and Methods section.
